# Contrasting Sequence with Structure:
# Pre-training Graph Representations with PLMs

**Louis Robinson**
InstaDeep

**Timothy Atkinson**
InstaDeep

**Liviu Copoiu**
InstaDeep

**Patrick Bordes**
InstaDeep

**Thomas Pierrot**[*]
InstaDeep

**Thomas D. Barrett**[*]
InstaDeep

{l.robinson,t.atkinson,l.copoiu,p.bordes,t.pierrot,t.barrett}@instadeep.com

## Abstract

Understanding protein function is vital for drug discovery, disease diagnosis, and protein engineering. While Protein Language Models (PLMs) pre-trained on vast protein sequence datasets have achieved remarkable success, equivalent Protein Structure Models (PSMs) remain underrepresented. We attribute this to the relative lack of high-confidence structural data and suitable pre-training objectives. In this context, we introduce BioCLIP, a contrastive learning framework that pre-trains PSMs by leveraging PLMs, generating meaningful per-residue and per-chain structural representations. When evaluated on tasks such as protein-protein interaction, Gene Ontology annotation, and Enzyme Commission number prediction, BioCLIP-trained PSMs consistently outperform models trained from scratch and further enhance performance when merged with sequence embeddings. Notably, BioCLIP approaches, or exceeds, specialized methods across all benchmarks using its singular pre-trained design. Our work addresses the challenges of obtaining quality structural data and designing self-supervised objectives, setting the stage for more comprehensive models of protein function. Source code is publicly available[2].

## 1 Introduction

The study of proteins, central to cellular function, impacts fields such as medicine, biotechnology, and computational biology. While the amino acid sequence of a protein carries vital information, its 3D structure often holds the key to understanding its function and putative interactions. Machine learning, especially Protein Language Models (PLMs), has recently revolutionized protein modelling. PLMs pre-trained on extensive sequence data have demonstrated the ability to capture intrinsic relationships in amino acid sequences [1] in rich representations that can be leveraged for various downstream applications [2] - mirroring the trends observed in other domains such as natural language processing [3, 4] and computer vision [5]. However, despite the success of machine learning in protein structure prediction, epitomized by AlphaFold [6], general pre-trained protein structure models (PSMs) have not yet found the same ubiquitous application as their sequence-based counterparts. We attribute this to two main challenges; (i) data scarcity and (ii) objective complexity.

High-quality protein structure data is hard to come by and often expensive. Methods such as X-ray crystallography and cryo-electron microscopy, though very insightful, are not without limitations

---

[*]Equal supervision.
[2]https://github.com/instadeepai/bioclip

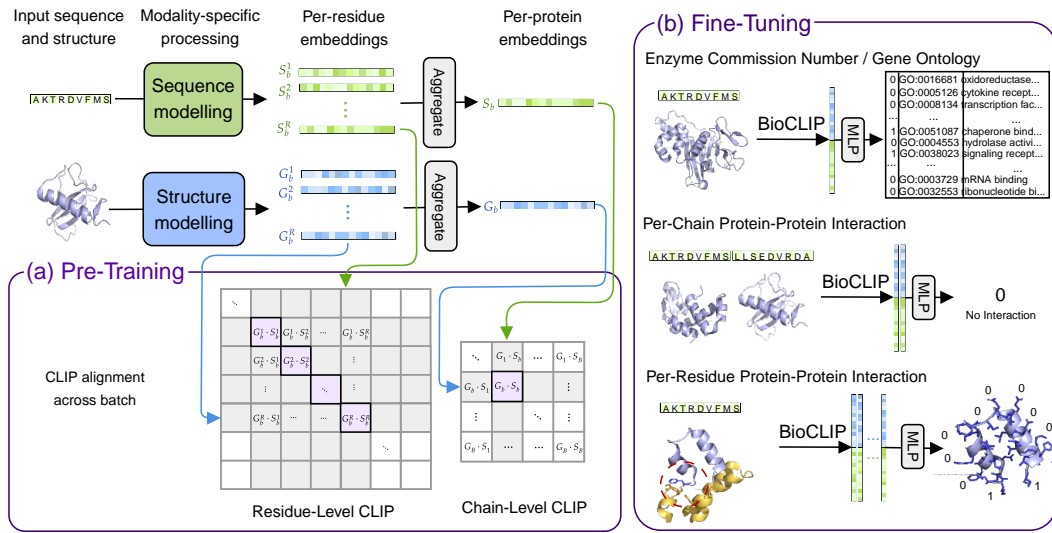

Figure 1: Pre-training, fine-tuning and downstream task illustrations. The modules in light blue are tuned, the modules in green are fixed.

[7]. Whilst recent tools such as AlphaFold and ESMFold have enabled the generation of massive protein structure datasets, they only predict the 3D coordinates, and can still be imperfect especially for multi-state proteins or proteins with shallow MSAs [8, 9]. However, objective formulation remains a significant hurdle. While masked sequence prediction has proven highly effective for pre-training PLMs, defining self-supervised objectives for structure data is far more challenging due to the continuous and multi-dimensional nature of protein structures.

Motivated by this, we introduce BioCLIP, a self-supervised contrastive learning framework for building latent representations of protein structures and sequences. The core intuition behind BioCLIP is to leverage the quality of pre-trained PLM embeddings trained on abundant sequence data to facilitate the training of a PSM. The model employs a loss function inspired by Contrastive Language–Image Pretraining (CLIP) [10], incorporating both per-residue and per-chain embeddings to create a comprehensive representation of protein structures.

We validate BioCLIP's efficacy through tests on protein-protein interaction prediction, GO-term annotation, and Enzyme Commission number prediction. Our findings underscore three points: (1) BioCLIP's pre-trained Graph Neural Network (GNN) surpasses conventional training methods, (2) structural embeddings enhance sequence embeddings and usually boost performance when combined, and (3) BioCLIP approaches or outperforms specialized methods.

## 2 Related Work

**Protein Language Models** Recent advances in scaleable transformer architectures [11] have facilitated an explosion in pre-trained language models, e.g. [12, 13, 4]. Correspondingly, a number of pre-trained *protein language models* have emerged, employing the same principles of auto-regressive or masked-prediction while targeting the vast available protein sequence data. For example, the ESM family of BERT-like models are trained to do masked prediction [14] which can be used for a variety of downstream tasks [15, 16, 17], ProtGPT2 is an autoregressive model capable of generating novel, realistic protein sequences [18]. While these PLMs in many ways represent a state-of-the-art, the inclusion of additional structural information may improve performance in practice [19].

**Protein Structure Models** Protein structure is a key modality in modelling of protein function, with a variety of research interest in both predicting protein structure [6, 20] and leveraging predicted or experimentally obtained structural information for protein modelling [21]. With the view that protein structure, represented as a set of residue coordinates in 3D space, can directly be mapped to a graph structure, there is a clear affinity with the subject of graph neural networks. In particular, a

number of GNN architectures have emerged in the past decade which specifically target invariance and equivariance with respect to the 3D coordinate system [22, 23, 24], some of which have shown promise in protein tasks such as prediction of solubility [25], function [26] and binding affinity [27]. However these *protein structure models* (PSMs) face bespoke challenges, such as the well-studied 'over-smoothing' problem for GNNs [28] that limits the size and depth of these models. Additionally, there is a relative lack of available structural data; consider that at the time of writing, the Protein Data Bank [29] has hundreds of thousands of experimentally obtained structures, in comparison to the millions of available protein sequences [30], although we note that this issue can increasingly be mitigated using predicted structures e.g. [31, 32].

**Pre-trained Structure Models**  Recently, a variety of techniques for contrastive learning on graph representations have been proposed. Typically, these incorporate some form of structural augmentation [33], structure masking [34] or network perturbation [35] to create neighborhoods of structure representations for contrastive learning. Of particular relevance to this work, GearNet [36] and Mol-CLR [37] propose graph-augmentation approaches to contrastive learning of protein and molecule structures, respectively. [38] propose a mask-prediction method where a GNN is trained to reconstruct pairwise distances and angles between residues. However, as these techniques focus on contrastive learning between graph structures, they are, in isolation, unable to leverage the vast available protein sequence data to improve their representations.

**Multi-modal Protein Embeddings**  As both sequence and structure are considered key modalities for modelling protein function, a number of works naturally consider combined representations for downstream tasks. For example, a number of works aims for a 'best of both worlds' by incorporating amino acid embeddings, obtained from a pre-trained PLM, as node features to a PSM that is then trained to predict protein function [39, 40]. However, these approaches do not incorporate unsupervised learning into their PSM components. The approach taken in this work can be motivated by [41], which presents experimental results showing that representations of sequences obtained via unsupervised learning, specifically with the ELMo model [42], are more effective in downstream tasks than hand-crafted representations. Further motivating the unsupervised learning of complimentary structural representations, [42] find that although the embeddings are obtained from sequence alone, they do not benefit from including hand-crafted structural representations. In contrast to these directions, a number of works provide avenues for multi-modal representations through large-scale contrastive learning, particularly in the case of image-text modalities [10, 43]. These methods provide a systematic way to build multi-modal representations of data, which we leverage here, alongside the ubiquitous success of PLMs, to achieve effective pre-training of data-scarce PSMs.

## 3   BioCLIP

BioCLIP is visualised in Figure 1. The core idea is to pass as input both a sequence and a structure representation of a given protein through a PLM and a PSM, respectively, to obtain per-residue and per-protein level embeddings. For a given sequence embedding $S_a$ and structure embedding $G_b$, the scaled cosine distance metric can be used to measure the similarity of the embeddings,

$$d\left(S_a, G_b\right) = \frac{S_a \cdot G_b}{||S_a|| \cdot ||G_b||}. \tag{1}$$

For a batch of sequence embeddings $S = \{S_1 \dots S_N\}$ and structure embeddings $G = \{G_1 \dots G_N\}$ where $S_i$ corresponds to $G_i$, a CLIP-style loss can be employed,

$$L_S = -\frac{1}{N} \sum_{a=1}^{N} \log \frac{e^{d(S_a, G_a)}}{\sum_{b=1}^{N} e^{d(S_a, G_b)}}, \tag{2}$$

such that by minimising the contrastive loss term for a given batch of proteins, the model is trained to produce aligned embeddings for paired sequence-structure inputs, which are far away from all other embeddings produced by the model. An equivalent method is used at the level of per-residue embeddings across all proteins in the batch. These learned embeddings can then be used for a variety of down-stream tasks; for example, for a given protein with sequence and structure representation,

the outputs of the PSM and the PLM can be concatenated and passed through an MLP to predict whether that protein has a given Enzyme Commission number.

In principle, while a separate BioCLIP model can be trained end-to-end at the level of both proteins and their component amino acids, these tasks are inherently related. Each structure or sequence is by definition a composition of its component residue nodes and amino acid types, respectively. There is a vast literature demonstrating that training a single backbone model for multiple, related, tasks typically yields better generalisation performance [44]. It is therefore beneficial to treat both sequence and amino-acid representations as heads of the same underlying sequence encoder, and similarly treat structure and residue representations as heads of the same underlying structure encoder. Then by minimising the sum of contrastive losses, we simultaneously minimise both the protein-level loss and the residue-level loss and benefit from the commonalities in the two problems.

We leverage the available massive pre-trained PLMs which already provide robust representations of the protein sequences, e.g. [45, 46, 18, 47, 48]. By appropriately choosing a pre-trained PLM, we are able to fix the sequence and amino-acid representations and then derive robust structural representations from limited sequence-structure pairs, reflecting the availability of substantially larger datasets of protein sequences. In our experiments, we use an instance of ESM [14] which is a state-of-the-art BERT-style PLM. The PSM used is a type of SE(3)-invariant graph neural network based on prior work [49]. By design, all node and edge features passed to the network are SE(3)-invariant, and any message passing applied on top of them inherently maintains this. Within the message passing mechanism, we use a type of graph attention network [50] with multi-head dot-product attention [11].

BioCLIP is pre-trained on a dataset of approximately 500,000 sequence-structure pairs obtained from the RCSB PDB databank [29]. Once the model has been pre-trained it can then be used for a variety of downstream tasks, such as those described in Section 4. Full details of the implementation used here are given in Appendix A.

## 4 Experiments

### 4.1 Tasks

To investigate whether BioCLIP is capable of learning meaningful structural representations that offer novel benefits on top of those already available from the underlying PLM, we empirically evaluate their performance when used as a basis for three downstream tasks. These tasks are visualised in Figure 1. Full details of the downstream tasks and their configurations are provided in Appendix B.

- **Function Prediction:** A binary protein classification task, based on datasets used in [26], where the goal is to predict enzyme-commission numbers and three gene ontology (GO) tasks: biological-process (BP), molecular-function (MF) and cellular component (CC).

- **Protein-Protein Interaction:** A binary classification task where, given two proteins, the objective is to predict whether or not they interact. We study the Human and S.cerevisiae tasks that are introduced in [40].

- **Per-Residue Protein-Protein Interaction:** A binary classification task where, given two biological sub-units within two distinct biological molecules, the objective is to predict whether or not they interact. This task is taken from [27].

### 4.2 Models

Across all tasks we utilise the same BioCLIP pre-trained GNN. During fine-tuning we take the structure representations obtained from the GNN *before* the application of a final MLP, such that initially, the structure representation differs from the sequence representation by a non-linear transformation. These structure embeddings are then concatenated with sequence embeddings from ESM and passed into an (initially untrained) predictor MLP model. The parameters of the GNN the final three layers of the ESM model and the final MLP are all fine-tuned for each specific task.

**Ablations**  To ablate BioCLIP's component parts and identify their contributions, we consider four variants of the method in fine-tuning:

- **GNN (random)**: The same GNN architecture as used in our BioCLIP pre-trained, but initialised with random weights. This variant is used as a control to measure the effectiveness of pre-training of the GNN. All parameters in the GNN and the final MLP are fine-tuned.

- **GNN (pretrained)**: The GNN architecture, pre-trained to align to the ESM model. This variant is used to measure the effectiveness of pre-training the structure model in isolation. Again, all parameters in the GNN and final MLP are fine-tuned.

- **ESM**: The PLM model used in pre-training, with its final three layers fine-tuned. This variant is used as a control to measure the effectiveness of introducing the BioCLIP-aligned GNN for downstream tasks. All parameters in the last three layers of the ESM model and the final MLP are fine-tuned. Note that we did experiment with different methods of fine-tuning but found that it made little difference to the final performance.

**Task-specific Baselines**   For each of the downstream tasks, we further identify a recent method that represents the state-of-the-art, or close, for that task:

- **DeepFRI [26] (GO-term tasks)**: This model represents the protein structure as contact maps, and employs a frozen protein language model to provide input node features for a three-layer graph convolutional network [51], which is pre-trained on 10 million Pfam protein sequences and uses an LSTM architecture with 512 hidden units. A sum operation is used to pool the per-residue representations into a single protein representation, which is finally passed through an MLP. The authors expand the training dataset by using homology models from SWISS-MODEL which they show boosts performance significantly, the dataset uses 30k non-redundant experimental PDB structures and 220k non-redundant homology models from SWISS-MODEL.

- **Jha et. al. [40] (protein-protein tasks)**: The authors use a GCN and a GAT to predict interactions between proteins. They use two pre-trained protein language models: SeqVec (LSTM) and ProtBert (Transformer) to obtain feature vectors for each residue. The SeqVec embedder produces a sequence representation by summing three representations: a 1-character convolution (CharNN), and bi-directional LSTM layers. The second PLM, ProtBert, is a BERT model is trained on the BFD-100 dataset [52], which has 2.1 million protein sequences. Finally, the per protein representation is obtained by averaging over residue activations. The PPI datasets used are from two organisms: Human and S. cerevisiae. The Pan's human dataset [53] is modified to remove duplicates and apply some filtering, see the original paper for details [40].

- **PeSTo [27] (per-residue protein interaction)**: The authors employ a deep GAT network on the atoms of a protein structure. Latent atom representations are obtained through 32 SO-3 invariant message-passing layers with increasing neighborhood sizes. Finally, cross attention is used from the atoms in each residue, to the corresponding residue.

## 4.3   Results

The results of the downstream evaluation experiments are summarised in Figure 2, with corresponding exact numerical values found in Appendix C. For each experimental configuration, we present results after 1, 2 and 3 epochs and also the final epoch from each fine-tuning experiment. From our experiments, we draw three important conclusions:

1. **BioCLIP's pre-trained structure representations are informative :** Observing the results for GNN (random) and GNN (pretrained), we find that for all 7 tasks and at every epoch, the pre-trained GNN provides better performance. This result is particularly stark in the Protein-Protein Interaction tasks, where the pre-trained GNN significantly outperforms the randomly initialised GNN in early epochs. From these observations, we conclude that pre-training a GNN via the BioCLIP method can result in meaningful representations of protein structure that can aid in downstream tasks.

2. **Aligned pre-trained structure and sequence representations are additive:** We find that in 25/28 cases, the results for fine-tuning of the full BioCLIP system outperforms the ESM model alone, and in 23/28 cases the pre-trained GNN alone. We argue that, although the GNN and ESM models are aligned, they provide different inductive biases which are mutually beneficial for fine-tuning. We note that in the cases where we did not observe a benefit in combining the representations, the drop in performance was very marginal.

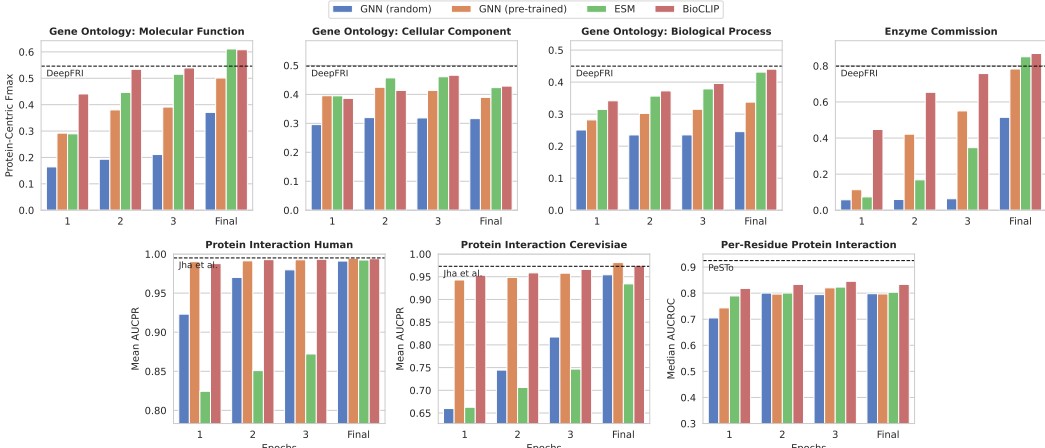

Figure 2: Performance metrics across 7 downstream tasks for: Random GNN, pre-trained GNN with the contrastive BioCLIP objective, ESM 150M with the last three layers tuned, and the BioCLIP model which combines ESM and the pre-trained GNN. For each task, we also provide a recent, task-specific, benchmark represented as a dashed line.

3. **BioCLIP is competitive with state-of-the-art methods:** In 4 of the 7 problems considered, we find that BioCLIP is able to out-perform or match a recent, state-of-the-art, method designed specifically for that task. In the GO: Molecular Function and Enzyme Commission problems, fine-tuning BioCLIP surpasses the results achieved by DeepFRI. Further, on the GO: Cellular Component and GO: Biological Process tasks, BioCLIP's performance reaches within 7% of the performance achieved by DeepFRI. For the two PPI tasks, we find that BioCLIP is able to match the performance of Jha et. al. Additionally, we find that BioCLIP is able to reach very close to the performance of Jha et. al. within only a few epochs. We also note that while BioCLIP struggled to reach the performance of PeSTo on per-residue protein interaction, it did outperform all of the methods that PeSTo was compared to in its original publication [27][3].

## 5   Conclusion & Future Work

This work introduces BioCLIP, a contrastive learning framework for learning structure and sequence representations of proteins, which we have demonstrated in a practical setting by aligning a GNN to a pre-trained PLM. We have carried out numerous empirical evaluations on a variety of downstream tasks for protein function prediction. We show that the representations derived from BioClip are meaningful, complementary to existing sequence embeddings and can be used to obtain competitive performance in comparison to task-specific methods. Overall, we believe that BioCLIP addresses the problem of limited structural data for pre-training in a systematic way, and provides a general template for models of protein function based on their full sequential and structural representations.

There are a number of areas for future work further developing the ideas presented here. Firstly, recent work [54] has demonstrated that a CLIP-style model can be effectively trained using a sigmoid loss, rather than a softmax loss, which may pave the way for training of BioCLIP with substantially larger batch sizes. An empirical investigation in this direction may yield improved performance on downstream tasks. We also note that the PSM used in this work was substantially smaller than the PLM, which may limit the richness of the structural embedding obtained. This reflects the broader problem within the geometric deep learning community of over smoothing in deeper networks [28]. Investigation into alternative models of structure, such as the EvoFormer module used in AlphaFold2 [6] and may therefore allow for larger, richer representations of structure.

---

[3]We performed a preliminary experiment pre-training a small version of the PeSTo architecture and fine-tuned it on the per-residue PPI task. As we found that we achieved similar performance to [27], we hypothesise that the difference in performance can be attributed to the PSM architecture and use of the all-atom representation.

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

# Appendix

## A    BioCLIP Implementation

### A.1    Loss

BioCLIP is inspired by the Contrastive Language-Image Pre-training (CLIP) method. Consider a batch of $n$ paired protein sequences and structures. The goal of BioCLIP is to learn a latent representation of each sequence, $S_i$, and structure $G_j$ so that the scaled cosine similarity,

$$d\left(S_i, G_j\right) = \frac{S_i \cdot G_j}{||S_i|| \cdot ||G_j||} \tag{3}$$

is maximised for paired sequences and structures, $i = j$, and minimised otherwise, $i \neq j$. This can be achieved by minimising the symmetric cross-entropy loss, with a term $L_S$ considering the discriminative power of the cosine similarity across all structures in the batch for each sequence, and similarly a term $L_G$ considering the discriminative power of the cosine similarity across all sequences in the batch for each structure,

$$L_S = -\frac{1}{n} \sum_{i=1}^{n} \log \frac{e^{\tau d(S_i, G_i)}}{\sum_{j=1}^{n} e^{\tau d(S_i, G_j)}}, \tag{4}$$

$$L_G = -\frac{1}{n} \sum_{i=1}^{n} \log \frac{e^{\tau d(S_i, G_i)}}{\sum_{j=1}^{n} e^{\tau d(S_j, G_i)}}, \tag{5}$$

where in both cases, $\tau \in \mathbb{R}^+$ is a learned 'temperature' controlling the sharpness of the softmax operator. Then the total 'protein-level' loss to be minimised is,

$$L_P = \frac{L_S + L_G}{2}. \tag{6}$$

Additionally, we are interested in learning aligned latent representations at a more granular level, as many downstream tasks consider per-residue function and interaction [26, 27]. Consider that each protein sequence is an ordered sequence of $m$ amino acids, $a_{i,1} \ldots a_{i,m}$, and, for each amino acid $a_{i,j}$ in that sequence, there exists a corresponding node $v_{i,j}$ within the structure's set of nodes $V_i = \{v_{i,1} \ldots v_{i,m}\}$. Then, as with the aligned protein-level representations $S_i, G_i$, we also desire aligned residue-level representations $S_i^j, G_i^j$ that, while conditioned on their global contexts $S_i, G_i$ are themselves latent representations of their corresponding amino acids and nodes. Then the symmetric cross-entropy loss is defined equivalently across the entire batch of residues,

$$L_A = -\frac{1}{n} \sum_{i=1}^{n} \frac{1}{m_i} \sum_{j=1}^{m_i} \log \frac{e^{\tau d\left(S_i^j, G_i^j\right)}}{\sum_{k=1}^{n} \sum_{l=1}^{m_k} e^{\tau d\left(S_i^j, G_k^l\right)}}, \tag{7}$$

$$L_V = -\frac{1}{n} \sum_{i=1}^{n} \frac{1}{m_i} \sum_{j=1}^{m_i} \log \frac{e^{\tau d\left(S_i^j, G_i^j\right)}}{\sum_{k=1}^{n} \sum_{l=1}^{m_k} e^{\tau d\left(S_k^l, G_i^j\right)}}, \tag{8}$$

and the total 'residue-level' loss to be minimised is,

$$L_R = \frac{L_A + L_V}{2}. \tag{9}$$

As we train a single model for both protein-level and residue-level contrastive learning, the overall loss term is simply,

$$L = L_P + L_R. \tag{10}$$

We did not find that it was necessary to introduce different weightings for the loss terms.

## A.2 Protein Language Model

We consider the `esm2_t30_150M_UR50D` instance of Evolutionary Scale Modeling (ESM) [47], which is a BERT-style [12] transformer architecture consisting of 30 layers and 150 million total parameters. We use the publicly available weights for this model, which are pre-trained to perform masked amino-acid prediction on 65 million unique protein sequences taken from the UniRef protein sequence database [55].

Each amino acid is represented as a token in an ordered protein sequence which is passed through a encoder-only transformer architecture. After processing by the ESM, the latent representation $S_i^j$ of each amino acid token $a_{i,j}$ is obtained from the final layer. The representation of the entire protein sequence $S_i$ is obtained by averaging over the amino acids' embeddings,

$$S_i = \frac{1}{m_i} \sum_{j=1}^{m_i} S_j^i \tag{11}$$

## A.3 Protein Structure Model

**SE(3)-invariance** We employ a SE(3)-invariant graph neural network based on prior work [49]. Each protein $p$ is represented as a graph $G = (V, E, f_V, f_E)$ where $V$ is the set of nodes, $E$ is the set of edges, $f_V \in \mathbb{R}^{|V| \times N}$ is the initial node features in dimension $N$ and $f^E \in \mathbb{R}^{|E| \times M}$ is the initial edge features in dimension $M$. The set of edges $E$ is constructed by taking each node's $k$-nearest neighbours in euclidean space, and each edge $e \in E$ has an associated source node $s(e) : E \rightarrow V$ and target node $t(e) : E \rightarrow V$. The neighborhood $\mathcal{N}_v \subseteq E$ of a node $v$ is therefore defined,

$$\forall v \in V, \mathcal{N}_v = \{e \in E \mid s(e) = v\} \tag{12}$$

The initial node features $f_V^1 \ldots f_V^{|V|}$ and edge features $f_E^1 \ldots f_E^{|E|}$ contain information about the local geometry, and are SE(3)-invariant. After this, any conventional message passing system can be employed without breaking this invariance. Every node/residue $v_i$ has a 3D coordinate $z_i$ – chosen as the coordinate of the $\alpha$-carbon atom – and a feature vector $f_V^i$. There are several possibilities to define the feature vector $f_V^i$ (these can be combined via concatenation). In our experiments, we choose the following:

- A one-hot encoding of the type of amino acid residue. This one-hot encoding serves as input of an embedding layer that is learned during the training phase.

- A sinusoidal encoding of sequence position. As in [56], the order of nodes is fixed to the corresponding sequence order. For the $i$-th residue, the positional encoding is $(\phi(i, 1)...\phi(i, D))$ where $D$ is the embedding size.

Edge features are defined following the same method as [49]. For each residue $v_i$, a local coordinate system is formed by (a) the unit vector $t_i$ pointing from the $\alpha$-carbon atom to the nitrogen atom, (b) the unit vector $u_i$ pointing from the $\alpha$-carbon to the carbon atom of the carboxyl (-CO-) and (c) the normal of the plane defined by $t_i$ and $u_i$: $n_i = \frac{u_i \times t_i}{\|u_i \times t_i\|}$. Finally, setting: $q_i = n_i \times u_i$, the edge features are then defined as the concatenation of the following:

- relative position edge features: $p_{j \rightarrow i} = \left( n_i^T u_i^T q_i^T \right) (x_j - x_i)$

- relative orientation edge features: $q_{j \rightarrow i} = \left( n_i^T u_i^T q_i^T \right) n_j$, $k_{j \rightarrow i} = \left( n_i^T u_i^T q_i^T \right) u_j$, $t_{j \rightarrow i} = \left( n_i^T u_i^T q_i^T \right) v_j$

- distance-based edge features, defined as radial basis functions: $f_{j \rightarrow i, r} = e^{-\frac{\|x_j - x_i\|^2}{2\sigma_r^2}}$, $r = 1, 2...R$ where $R = 15$ and $\sigma_r = 1.5$

**Message Passing**  We use a type of graph attention network [50] described as follows. Consider hidden node features at layer $l$, $x_l^1 \ldots x_l^{|V|}$ where for $l = 0$, these correspond to the initial SE(3)-invariant node features e.g. $x_0^v = f_V^v, \forall v \in V$. Then each hidden node representation undergoes layer normalisation with affine parameters [57],

$$\forall v \in V, y_l^v = \texttt{layer\_norm}\left(x_{l-1}^v\right), \tag{13}$$

For node $v$'s neighborhood $\mathcal{N}_v$, we construct a representation of each outgoing edge $e$ by concatenating its original features with the normalised hidden features of the edge's target node $t(e)$,

$$\forall e \in E, x_l^e = f_E^e \parallel y_l^{t(e)}, \tag{14}$$

where $a \parallel b$ denotes concatenation of vectors $a$ and $b$. Multi-head dot-product attention [11] with $H$ heads is applied over the neighborhood, where the output of the $h$th head is computed,

$$n_{l,h}^v = \sum_{e \in \mathcal{N}_v} \frac{\exp\left(W_{l,h}^q y_l^v \cdot W_{l,h}^k x_l^e\right)}{\sum_{e' \in \mathcal{N}_v} \exp\left(W_{l,h}^q y_l^v \cdot W_{l,h}^k x_l^{e'}\right)} W_{l,h}^x x_l^e, \tag{15}$$

and $W_{l,h}^q, W_{l,h}^k, W_{l,h}^x$ are the learned query, key and value projections, respectively. Finally, a residual connection [58] is used to ensure effective gradient propagation,

$$\forall v \in V, x_l^v = \phi_l(n_l^v) + x_{l-1}^v \tag{16}$$

where $\phi$ is a simple MLP with a single hidden layer and layer normalisation applied at its input, and $n_l^v$ is obtained by concatenating the outputs of the $H$ heads, $n_{l,1}^v \ldots n_{l,H}^v$.

**Latent Representation**  After $L$ steps of message passing, 2-layer MLP $\phi_v$ is used to map each node/residue into the PLM embedding dimension to obtain the per-residue representation,

$$\forall v \in V, G^v = \phi_v\left(x_L^v\right) \tag{17}$$

To obtain the representation of the structure, we employ cross attention mechanism to reduce the nodes' representations into a single vector. A single learned vector takes the role of the query $W_{l,h}^q y_l^v$ in equation 15, giving a reduced representation of the structure after $L$ layers of message passing, $G_L$. Finally, an MLP $\phi_S$ with two hidden layers, is used to map the reduced representation to the PLM embedding dimension to obtain the per-structure representation,

$$G = \phi_S(G_L) \tag{18}$$

### A.4  Pre-training

**Data**  The BioCLIP GNN is pre-trained by processing all mmcif files in the Protein Data Bank (PDB) [29] with a cutoff before 2020-05-14, resolution $< 9\text{Å}$, and no single amino acid accounting for more than 80% of the sequence [59]. This results in around 490 k structures.

**Dataloader**  We found we could get a boost in performance by sampling the batch based on sequence similarity. Specifically, we hypothesised that the majority of protein chains will be completely different and very easy to satisfy the pre-training loss. To force the batch to have more similar sequences we clustered our pre-training data at 50% and minimum sequence coverage of 0.8, then we sample clusters for the batch proportional to the square root of the cluster size, and take four samples uniformly at random within each cluster.

**Hyperparameters**  The BioCLIP GNN is trained for 150 thousand steps with a batch size of 128 protein sequences. A loss is computed across representations $S_i, G_j$ for all 128 proteins in the batch. We consider all residues in the batch, and sample 2048 of them uniformly at random, without replacement, for the purposes of approximating the loss for contrasting amino acid-node representations $a_{i,j}, v_{k,l}$. We use the Adam optimiser [60] with a learning rate of $10^{-4}$ and default values for $\beta_1, \beta_2$. The GNN model has 3 layers of message passing with a hidden node representation dimension of 512. Each GAT layer is followed by an MLP which has input and output of size 512 and one hidden layer of size 2048. The per-chain and per-residue structure representation has an MLP which input size 512, two hidden layers of size 1024 and output size 640 to match ESM. All MLPs have the ReLU activation function.

## A.5   Fine-tuning

The initialisation of the full BioCLIP model for fine-tuning requires loading the pre-trained GNN parameters but discarding the final MLP, concatenating with the ESM embeddings then adding a randomly initialised MLP. This MLP has two hidden layers of size $h + c$ and an output size of $c$ which is the number of categories for that task, $h$ is the hidden size of the GNN, $h = 512$. We use the per-residue or per-chain representation depending on the task. For each task the batch size is 32 and we use the adamw optimiser with learning rate and weight decay equal to $10^{-4}$ (weight decay is 0 for the per-residue PPI task). We train GO/EC for 20 epochs, PPI for 30 epochs and per-residue PPI for 5 epochs.

## B   Downstream Tasks

**Function Prediction**  To evaluate BioCLIP in the context of protein function prediction, we consider four multiple binary classification tasks: enzyme-commission numbers (EC) and three gene ontology (GO) tasks: biological-process (BP), molecular-function (MF) and cellular component (CC). All datasets are the same as in [26] and, in the case of comparison to DeepFRI, we recompute all methods' performances on identical test sets. The datasets are divided as follows: train/validation/test datasets are made up of 27581 / 3061 / 2991 examples for GO tasks and 15035 / 1665 / 1840 for the EC task. The number of terms in each task is as follows: BP 1943, MF 489, CC 320 and EC 538. Class imbalance is quite severe, the median positive class percentage across terms is BP 0.122%, CC 0.116% MF 0.105% and EC 0.057%. As each task is itself a collection of binary classification tasks, the loss function used is binary cross entropy, averaged across all terms. We evaluate performance with two criteria: (1) for a given term, how well does the model classify the proteins which have that term; (2) for a given protein, how good is the model at classifying which terms are positive. The reason we opted for two different metrics is due to the convoluted design of the GO nomenclature, where one protein sequence can have multiple GO terms. With the above mentioned metrics we will asses (1) which GO terms are easier to predict and (2) the degree to which we can comprehensively annotate test sequences.

**Protein-Protein Interaction**  We evaluate the learned BioCLIP representations on two benchmarks of protein-protein interaction introduced in [40]. The benchmarks Human and S.cerevisiae are composed of pairs of proteins annotated with a 0/1 label indicating whether proteins interact. The dataset contains positive protein-protein interaction pairs from the human protein reference database (HPRD) and the Database of Interacting Proteins (DIP) for humans and S. cerevisiae, respectively. After preprocessing steps, such as eliminating duplicates and removing proteins with fewer than 50 amino acids, the dataset has 37K and 17K interacting pairs for humans and S. cerevisiae. Negative instances, representing non-interacting protein pairs, are generated based on subcellular localization differences, with totals of 36,323 and 48,594 pairs for each organism, and homologous pairs are filtered using the CD-HIT tool at a 40% sequence identity cutoff. In total we have 22K in human and 9K for cerevisiae after filtering lengths below 1024. The datasets are relatively balanced: for the human targets there is 73% in the positive class, for cerevisiae there is 50%. The loss function in both tasks is cross entropy. We use the same metric as our baseline for this task [40] to be able to directly compare to BioCLIP. AUCPR is a reasonable metric for this task as it is robust to the relatively significant class in-balance that is present.

**Per-Residue Protein-Protein Interaction** A formulation of a per-residue protein interaction task is provided in [27]. The task is as follows: given a static biological molecule extracted from a PDB file for instance a protein, DNA or RNA molecule, predict for all biological sub-units (residue, nucleotide, ligand, lipid or ion) the probability of interacting with another type of sub-unit which is not part of the same molecule. The data is constructed by parsing an assembly in a PDB file and, using a representative coordinate for all sub-units considered; over all sub-unit pairs belonging to different molecules, store the pairs within a distance cutoff, in this case 5Å. We use the data processing script provided with the original paper, which processes the following unique biological sub-units: 20 amino acids, 8 nucleic acids, 16 common ions, 31 ligands, and 4 lipids which amounts to 79 unique types of molecules. These are subsequently grouped into five groups: amino-acids, nucleotides, ions, ligands and lipids. As in [27] we evaluate on the MaSIF-site dataset [61, 62]. The authors use a model which can take a protein as input and for each residue produce five predicted probabilities for interaction with all five sub-unit types. Note that a given residue can be in contact with multiple other sub-unit types, or none at all. The loss function for this task is binary cross-entropy which has a positive class weighting as in the original paper. For comparison reasons, we use the same metric as PeSTo [27] which is median ROCAUC. Of the 27M residues in the training dataset $14\%$ of them are in contact with another residue (nucleotides is $0.64\%$, ion $1.77\%$, ligand $3.40\%$, lipid $0.06\%$). Since the residue-residue interactions are relatively balanced, we believe it is acceptable to use ROCAUC; which also has the advantage of always knowing what a random guess is, over AUCPR.

# C    Results Table

Exact numerical values corresponding to bar charts shown in Figure 2.

| Task | ESM | GNN (random) | GNN (pre-trained) | BioCLIP | SOTA | Epochs |
|---|---|---|---|---|---|---|
| MF | 0.289 | 0.164 | 0.292 | 0.440 | | 1 |
| | 0.446 | 0.193 | 0.380 | 0.533 | | 2 |
| | 0.515 | 0.210 | 0.390 | 0.539 | | 3 |
| | 0.611 | 0.370 | 0.500 | **0.607** | 0.546 | Final |
| CC | 0.394 | 0.295 | 0.395 | 0.386 | | 1 |
| | 0.456 | 0.319 | 0.424 | 0.413 | | 2 |
| | 0.460 | 0.318 | 0.413 | 0.466 | | 3 |
| | 0.423 | 0.316 | 0.389 | 0.428 | **0.497** | Final |
| BP | 0.314 | 0.250 | 0.281 | 0.341 | | 1 |
| | 0.356 | 0.234 | 0.302 | 0.372 | | 2 |
| | 0.378 | 0.235 | 0.315 | 0.395 | | 3 |
| | 0.431 | 0.245 | 0.337 | 0.440 | **0.449** | Final |
| EC | 0.073 | 0.057 | 0.113 | 0.447 | | 1 |
| | 0.168 | 0.059 | 0.421 | 0.653 | | 2 |
| | 0.346 | 0.063 | 0.550 | 0.757 | | 3 |
| | 0.850 | 0.514 | 0.782 | **0.868** | 0.798 | Final |
| PPI-Human | 0.824 | 0.923 | 0.990 | 0.987 | | 1 |
| | 0.850 | 0.970 | 0.991 | 0.992 | | 2 |
| | 0.872 | 0.979 | 0.992 | 0.993 | | 3 |
| | 0.992 | 0.991 | 0.994 | 0.993 | **0.995** | Final |
| PPI-Cerevisiae | 0.662 | 0.659 | 0.942 | 0.953 | | 1 |
| | 0.706 | 0.744 | 0.948 | 0.958 | | 2 |
| | 0.746 | 0.817 | 0.957 | 0.966 | | 3 |
| | 0.934 | 0.954 | **0.981** | 0.974 | 0.973 | Final |
| Res-PPI | 0.789 | 0.704 | 0.743 | 0.818 | | 1 |
| | 0.800 | 0.800 | 0.796 | 0.833 | | 2 |
| | 0.823 | 0.794 | 0.820 | 0.845 | | 3 |
| | 0.803 | 0.797 | 0.796 | 0.833 | **0.924** | Final |

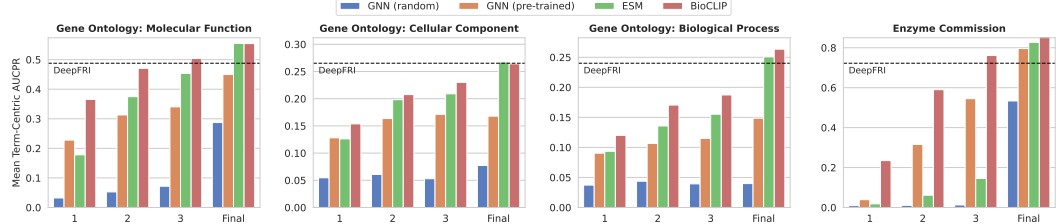

Figure 3: Performance with the alternative metric mean term-centric Fmax across the GO/EC downstream tasks for: Random GNN, pre-trained GNN with the contrastive BioCLIP objective, ESM 150M with the last three layers tuned, and the BioCLIP model which combines ESM and the pre-trained GNN. For each task, we also provide a recent, task-specific, benchmark represented as a dashed line.

## D   Results Table

Exact numerical values corresponding to bar charts shown in Figure 3.

| Task | ESM | GNN (random) | GNN (pre-trained) | BioCLIP | SOTA | Epochs |
|------|------|------|------|------|------|------|
| MF | 0.177 | 0.032 | 0.227 | 0.365 | | 1 |
| | 0.374 | 0.052 | 0.312 | 0.470 | | 2 |
| | 0.453 | 0.071 | 0.340 | 0.503 | | 3 |
| | 0.554 | 0.287 | 0.449 | **0.554** | 0.487 | Final |
| CC | 0.126 | 0.054 | 0.128 | 0.153 | | 1 |
| | 0.198 | 0.060 | 0.163 | 0.207 | | 2 |
| | 0.208 | 0.052 | 0.171 | 0.230 | | 3 |
| | **0.267** | 0.077 | 0.167 | 0.264 | 0.265 | Final |
| BP | 0.093 | 0.037 | 0.090 | 0.119 | | 1 |
| | 0.135 | 0.043 | 0.106 | 0.170 | | 2 |
| | 0.155 | 0.039 | 0.114 | 0.187 | | 3 |
| | 0.250 | 0.039 | 0.148 | **0.263** | 0.240 | Final |
| EC | 0.018 | 0.009 | 0.039 | 0.234 | | 1 |
| | 0.061 | 0.010 | 0.315 | 0.589 | | 2 |
| | 0.145 | 0.012 | 0.544 | 0.760 | | 3 |
| | 0.826 | 0.533 | 0.795 | **0.854** | 0.722 | Final |

