# OpenReview forum: "Contrasting Sequence with Structure: Pre-training Graph Representations with PLMs"
_NeurIPS.cc/2023/Workshop/AI4Science — NeurIPS2023-AI4Science Poster_

### Official Review · Reviewer_KSmV · 2023-10-20
**protein sequence and structure co-pretrain**

**Rating:** 7
**Confidence:** 3

**Review:**

This paper leverage CLIP strategy to pretrain a network for both protein sequence and structure. Generated representations show on par performance with SOTA methods on selected protein function prediction task and protein- and residue-level interaction prediction task. This paper is overall clear and easy to read. Although there is no major improvement of performance, considering it's a foundation which could be scalable for larger datasets and finetuned for specific tasks, I see its great potential to be further optimized and extended, for example on high-quality predicted structures.
One concern I have is why a small ESM model is used. I suggest at least try the 650M one. Also there seems to be no weighting across different loss terms. Is it a decision after investigation of scale of losses?

Minor issue:
Line 214: Rephrase 'the best BioCLIP performance is within 7% of the performance achieved by DeepFRI'.